# The impact of the COVID-19 pandemic on the mental health of families dealing with attention-deficit hyperactivity disorder

Alexis Winfield©, Carly Sugar©, Barbara Fenesi®*

Faculty of Education, Western University, London, Ontario, Canada

© These authors contributed equally to this work.
* bfenesi@uwo.ca

**Data Availability Statement:** There are ethical considerations surrounding sharing of interview-based data. Given the qualitative interview-based nature of the data, there is personal, detailed, and sensitive information contained within the

## Abstract

### Background

The COVID-19 pandemic uprooted regular routines forcing many children to learn from home, requiring many adults to work from home, and cutting families off from support outside the home. Public health restrictions associated with the pandemic caused widespread psychological distress including depression and anxiety, increased fear, panic, and stress. These trends are particularly concerning for families raising neuroatypical children such as those with Attention-Deficit Hyperactivity Disorder (ADHD), as these children are already more likely than their typically developing peers to experience comorbid mental health issues, and to experience greater distress when required to stay indoors. Families with children who have ADHD are also at greater risk for experiencing heightened familial stress due to the challenges associated with managing ADHD behavioural symptoms, greater parental discord and divorce, and greater financial difficulties compared to other families. The current study engaged families comprised of at least one child diagnosed with ADHD to elucidate 1) the unique ways that the COVID-19 pandemic affected their mental health and 2) the specific barriers these families faced to maintaining optimal mental wellbeing.

### Methods and findings

A total of 33 participants (15 parent-child dyads) engaged in virtual interviews. Content analysis revealed that the most frequently identified mental health effects for families were increased child anxiety and disconnectedness, as well as deteriorating parental mental health. The most frequently identified barriers to maintaining optimal mental wellbeing were lack of routine, lack of social interaction and social supports, and uncertainty and fear.

### Conclusions

Findings underscore areas of need during times of large-scale social isolation, specifically for families with children who have ADHD. This work contributes to a growing body of research aimed at creating safeguards to support mental wellbeing for vulnerable families during times of crisis.

transcripts that does not lend itself to ubiquitous public access. Although the interview data can be de-identified, they cannot be redacted in a way to make the content completely de-identifiable (e.g., the sharing of personal experiences and stories could potentially be recognized by someone). Redaction could also potentially remove vital data for replication. Western University's Research Ethics Board has imposed data sharing restrictions as they do not allow identifiable or de-identified data to be electronically transmitted outside of Western including open access repositories. Requests can be sent to Western University's Office of Human Research Ethics (Phone: 519.661.3036; Fax: 519.850.2466; email: wrem@uwo.ca). Requests can also be sent to the corresponding author (bfenesi@uwo.ca).

**Funding:** The author(s) received no specific funding for this work.

**Competing interests:** The authors have declared that no competing interests exist.

## Introduction

The Novel Coronavirus Disease (COVID-19) has caused substantial morbidity and mortality worldwide [1]. In March 2020, the world entered a state of lockdown to slow the spread of the virus. Schools and businesses were closed, and people were required to stay home and socially isolate from other households. COVID-19 and the associated public health restrictions caused widespread psychological distress including depression and anxiety, increased fear, panic, and stress among younger and older adults [2–6], children and adolescents [7,8]. This is particularly concerning for families with children who have Attention-Deficit Hyperactivity Disorder (ADHD), as these children are already more likely than their typically developing peers to experience comorbid mental health issues [9–11], and to experience greater distress when required to stay indoors [12]. Families with children who have ADHD are also at a greater risk for experiencing heightened familial stress [13–18] which often exacerbates ADHD symptoms and further amplifies familial stress. The current study aimed to identify the impact of the COVID-19 pandemic on the mental health of families with children who have Attention-Deficit Hyperactivity Disorder (ADHD).

COVID-19 has had an unprecedented impact on the mental and physical wellbeing of individuals all over the world [3]. Due to a dearth of knowledge surrounding the coronavirus at the pandemic's inception, the fear of the unknown alone exacerbated stress, depression, and anxiety in adults [2,6,19]. The social isolation due to the stay-at-home orders increased loneliness and boredom [20,21], which if left unchecked can have long-term consequences for physical and mental wellbeing including poor sleep quality, reduced cardiorespiratory health, decreased executive functioning, enhanced threat arousal, poor decision making and snowballing anxiety [4]. Participation in physical activity behaviours also declined, contributing to diminishing physical and mental health [22–25].

Among children, school closures left them disconnected from friends and peers [26,27] stunting their social connection and development [28] and removing vital emotional and academic support from teachers [29]. School closures have been associated with an array of health risk behaviours such as socio-emotional complications and reduced physical activity behaviours [29,30]. An increase in parental stress due to job loss and heightened parental workload within the home also increased the likelihood of child maltreatment [31,32]. Critically, children are especially susceptible to negative long-term health outcomes due to impoverished social, emotional, and physical environments [33,34]. With the pandemic presenting such unique and profound changes to these environments, it will be crucial to continue unpacking the long-term effects on these children's overall wellbeing.

During the height of the pandemic, families received very little external support. Parents took on an unprecedented workload, maintaining their jobs, stepping in as their children's teacher, and providing emotional support for their children simultaneously. An accumulation of stressors related to the pandemic has been associated with higher parental stress which negatively impacts parent-child interactions [31,32,35]. The stressors associated with the COVID-19 pandemic also made existing challenges such as mental health struggles and financial stressors even more difficult to manage [36].

### Research gap and objectives

Promisingly, more research is emerging that captures the unique experiences of families dealing with ADHD during the pandemic [37–42]. A recent systematic review on the impact of COVID-19 on the mental health outcomes of individuals with ADHD found that the pandemic exacerbated existing mental health and social difficulties including anxiety, disturbed sleep [43] and poor social connections [44]. Compared to neurotypical controls, individuals

with ADHD were also reported to have significantly worse conduct issues, which became magnified during the pandemic [45]. A consistent finding in the research underscores that individuals with ADHD have pre-existing mental and physical health challenges that the pandemic amplified [37–42,44–46]. Parents of individuals with ADHD also reported a dramatic decline in their own well-being during the pandemic [46] given the additional taxation on their children and on the family system as a whole. Students with ADHD also had greater challenges learning in an online environment than their typically developing peers, leading to greater feelings of distress, loneliness, and poorer self-efficacy in the academic domain—an area in which students with ADHD already often struggle [47]. The level of academic support offered by parents during homeschooling to their children with ADHD also varied according to the child's gender, with males receiving less support [48]. The imbalance of parental academic support towards males with ADHD during the pandemic may be related to an increase in their conduct issues during this time, but the directionality of this relationship remains unclear. The theme of increased loneliness during the pandemic among children with ADHD has been supported in other work as well [49]. Importantly, a consistent finding among this research is that there are distinct mental health challenges that persist for longer among families dealing with ADHD. The current study complements existing work by offering a novel qualitative lens to give voice and nuance to the experiences of families which is often left uncaptured by quantitative research [50,51]. It is vital to capture how the pandemic has uniquely impacted families with children who have ADHD to better understand the necessary safeguards for mental well-being among vulnerable families during times of crisis. Thus, the current study explored how the COVID-19 pandemic 1) impacted the mental health of families with children who have ADHD, and 2) created barriers to maintaining optimal mental health for families with children who have ADHD. Overall, this study advances our understanding of the environmental and social impact of COVID-19 on the mental wellbeing of families with children with ADHD.

## Study rationale and theoretical framework

While the pandemic has had a substantial impact on the lives of all individuals and families, there is a particular need to understand the impact of the pandemic on high-needs households, such as on families with children who have ADHD. Systems Theory [52–54] helps explain why examining the impact of the pandemic on families dealing with ADHD is especially important by contextualizing the functioning of various "systems" using six component elements. Importantly, the focus of the current study was the family system. The first component element in the family system is the "system" itself, which is comprised of interrelated parts such as the guardians, children, and extended family. The second component element is the larger "complex system" that the family system resides within, such as the broader community, which is comprised of school, friends, peers, places of employment, colleagues, sports teams (coaches, teammates), therapy groups, and many other parts. The third component element is the "ecological system", which influences behaviour and is often composed of elements from the family and complex systems. The fourth component element is homeostasis, which is a steady state of conditions that the system is always seeking to maintain. The fifth component element is adaptation and is the process by which a system changes its behaviours to adapt to new environmental factors and return to homeostasis. And lastly, the sixth component element is a feedback loop, where the output of a system ultimately impacts the input, causing the system to circularly feed back into itself.

Systems Theory emphasizes a breakdown in component elements as a key contributing factor in the deterioration of mental and physical health among families during COVID-19

[2,3,6,19–21]. Specifically, the complex and ecological systems were massively fractured as families were cut off from the broader community and forced into social isolation. The family system's ability to maintain homeostasis was compromised as this component depends on the successful interrelation with the complex and ecological systems e.g., school, friends, family, caregivers, extracurriculars, therapy groups, etc. Given the lack of access to supports outside the home, families struggled to adapt to the new environmental pressures of social isolation. Systems Theory also points to potentially unique struggles among families dealing with ADHD compared to typical families during the pandemic, as their abilities to maintain homeostasis, adapt to challenging circumstances, and to maintain a positive feedback loop are all likely more dysfunctional to begin with. Regarding homeostasis and adaptability, families with children who have ADHD experience higher baseline familial stress due to the challenges managing ADHD behavioural symptoms [14,15], greater parental discord and divorce [13,18] and greater financial difficulties compared to other families [16,17]. Although what constitutes homeostasis can be different across families, it is evident that families with ADHD experience more challenging life circumstances than typical families, likely creating environments that separate these families from their homeostatic balance more often. Regarding the feedback loop, research has shown that the major outputs of the family system during COVID-19 were increased mental health issues among both parents and children, and that these were worse among those dealing with ADHD. Indeed, families dealing with adolescent ADHD [37] and special needs [38] were shown to have sustained negative mental health functioning even after the stay-at-home orders were lifted compared to typical families. The resulting input back into the family system was worse parent-child interactions [12], which has also been shown to be more severe in family dynamics where there are children with special needs [38,55,56]. Thus, the magnitude of the negative effects of the pandemic on the mental health of families dealing with ADHD is likely to be greater given the increased challenges maintaining homeostasis and adapting to environmental stressors, as well as a feedback loop burdened by a vicious cycle of worsening familial mental health and family dynamics.

Attachment Theory is another framework that underscores the importance of understanding how COVID-19 impacted families dealing with ADHD. Attachment Theory posits that children who have attentive and responsive caregivers develop a critical sense of security (secure attachment) [57,58]. In contrast, children who experience inattentive and unresponsive caregivers develop a lack of trust and fail to develop a secure base (insecure attachment). Attachment styles form the basis for how individuals respond to their environment, especially to stressful situations [59]. COVID-19 and Attachment Theory are intricately linked, as high stress situations tend to activate attachment systems [60,61]. For example, children with a secure attachment style are better able to handle stressful situations as they have established connections with others and healthy levels of self-esteem and self-worth. However, children with an insecure attachment style often respond maladaptively to stressful situations as they do not have responsive caregivers to help foster the resilience required during hardship. There has been extensive work highlighting how children with ADHD often exhibit insecure attachment styles [62–65]. The genesis of this relational interaction is a chicken-and-egg problem. Was it inattentive and stressed parents that were not able to meet the child's needs, or the child's challenging behaviour that made it difficult for bonding to occur that led to the insecure attachment dynamic? Or perhaps a combination of both? Regardless, recent work has shown that children and adults with insecure attachment styles experienced the COVID-19 pandemic significantly worse than their securely attached counterparts [60,66]. Specifically, insecurely attached individuals experienced increased anxiety, depression, and emotional dysregulation during the pandemic, and were more likely to struggle with the effects of social isolation e.g., disconnectedness. Given that children with ADHD often present with insecure attachment

[62–66], as do their caregivers [67], the maladaptive responses to the pandemic were likely worse for families dealing with ADHD. Thus, the current study aimed to elucidate how the COVID-19 pandemic affected the mental health of families dealing with ADHD and the specific barriers these families faced to maintaining optimal mental wellbeing.

## Methods

### Participants

Parents of children with ADHD who previously participated in studies with the researchers' lab, and provided contact information for future studies, were contacted and recruited for the present study. Recruitment occurred between October 2020 and January 2021. The institution's Cognitive Neuroscience Research Registry was also utilized to recruit participants. Additionally, snowball sampling was used by asking participants if they knew of any other families with a child with ADHD who would be interested in participating.

A total of 33 participants from Ontario, Canada took part in the study, with 15 independent family units. A total of 25 family units were contacted, for a response rate of 60%. To be eligible for the study, parent participants must have had at least one child with an ADHD diagnosis who was living with them for at least some of the time during the COVID-19 pandemic. The wording "for at least some of the time" was used to be inclusive of families with shared custody arrangements. Parents and children provided informed written consent and assent, respectively.

### Procedure

Prior to engaging in the interviews, participants (parents and their children) completed consent and assent forms via email, and parents completed an online demographics survey through university approved software. Following, participants engaged virtually in a semi-structured interview with a researcher via Zoom. The interviews lasted approximately one hour for the parent(s) interview and 30 minutes for the child interview. Both parent and child participants were compensated for their time in the form of a $10 Amazon gift card.

### Materials

**Online survey.** Parent participants completed an online demographics questionnaire which included questions about age, gender, income, race, and education. Specific questions were also asked about the child participant's ADHD diagnosis (see Table 1).

**Semi-structured interviews.** The Social Ecological Model (SEM) was used to design the interview questions. The SEM is often used in research designed to identify barriers to behaviour by acknowledging the interdependences between intrapersonal, interpersonal, institutional, community and policy factors influencing behaviour [68]. The verbiage "barriers" was used throughout the interview guide, with specific questions targeting the different interrelated levels (e.g., asking about intrapersonal, interpersonal, community factors impacting mental wellbeing for both children and parents).

The parental interview was comprised of two parts. The first part asked parents about the unique barriers for their children during the pandemic due to their ADHD diagnosis, and whether there had been changes in the presenting symptoms associated with their child's ADHD. The second part focused on questions related to mental health; the questions aimed to gather information about the parents' mental health and their child's mental health before and during the pandemic, and to gain insight into the barriers they experienced to maintaining mental wellbeing. The full interview guide is provided as supplementary material (S1 Appendix).

**Table 1. Demographic characteristics.**

| Demographic characteristic | Frequency |
|---|---|
| Gender of parent | |
| • Male | 2 |
| • Female | 13 |
| Gender of child | |
| • Male | 12 |
| • Female | 6 |
| Age of child (yrs) (Mean / SD) | 10.16/2.2 |
| Race of parent | |
| • Caucasian | 14 |
| • Black | 1 |
| Education | |
| • Some post-secondary | 1 |
| • Post-secondary | 4 |
| • University/Professional Degree | 7 |
| Household income | |
| • 20,000–30,000 | 1 |
| • 80,000–90,000 | 1 |
| • 90,000–100,000 | 2 |
| • 100,000+ | 5 |
| • Prefer not to say | 6 |
| Comorbid neurological diagnosis (child) | 3 |
| Comorbid mental disorder diagnosis (child) | 3 |
| Comorbid physical/auditory/visual disorder diagnosis | 4 |

The child interview was also comprised of two parts. The first part asked for demographic information of age and gender. The second part asked three questions: 1) How do you feel about the COVID-19 pandemic?; 2) What is the hardest part about dealing with the COVID-19 pandemic?; and 3) What has been the biggest change for you since the COVID-19 pandemic? Children were interviewed separately from their parents to offer them full freedom of expression, as some children may be reluctant to share some thoughts with their parents present. Children and parents were asked age-appropriate questions that were different, but related, to account for possible differences in communication or self-reflection ability.

## Qualitative data analysis

A post-positivist paradigm was used in data collection and analysis [69]. Given that the current study aimed to understand experiences related to the COVID-19 pandemic, which the researchers were also experiencing themselves, a post-positive approach was most appropriate to acknowledge researcher bias and potential influences of personal experience and background knowledge related to the pandemic [70]. Inductive content analysis was used to analyze interview responses [71]. Content analysis compresses texts into content categories based on explicit codes and was used to examine trends and patterns in transcribed audio-recorded interviews [72,73]. The technique was inductive as content categories were derived from the data. The content analysis aimed to identify, analyze, and report common themes from both parent and child interview transcripts. Specifically, the analysis aimed to identify the effects that the pandemic had on the mental health of families with children with ADHD, as well as the unique barriers they faced to maintaining optimal mental health.

In the first step of the data analysis process, digital recordings of interviews were transcribed verbatim using a professional transcription service. The transcribed interviews were then read over independently by two researchers to help them become familiar with the data.

Following a familiarization phase, the two researchers collaboratively generated a preliminary codebook to categorize interview content into meaningful groups based on emerging themes. The preliminary codebook was then applied to all transcripts. Following, researchers consulted on the success of the codebook in capturing key themes in the data. Any coding discrepancies were reviewed and discussed, and final coding decisions were made. A final codebook was generated and reapplied to all transcripts. All themes and subthemes are reported in tables below; the three most frequently identified themes and their corresponding most frequently identified four subthemes are discussed. Excerpts from participants are provided to illustrate key findings.

**Trustworthiness.**   Multiple measures were implemented to ensure trustworthiness by considering credibility, dependability, and transparency. To ensure credibility, investigator triangulation was used. Multiple investigators took part in the research process, particularly in the data analysis stage. This contributed to credibility by confirming findings across multiple investigators and by minimizing any research bias [74,75]. Dependability was promoted by creating explicit and repeatable methods through the use of recruitment scripts and interview guides. Detailed methodology and research processes were recorded throughout the research process. Western University's Non-Medical Research Ethics Board approved the study (protocol #116190).

## Results

The current study explored how the COVID-19 pandemic 1) impacted the mental health of families with children who have ADHD, and 2) created barriers to maintaining optimal mental health for families with children who have ADHD. The results are divided into two sections, each representing its corresponding research objective. Both research objectives are represented as a topic theme with subthemes generated from data analysis. Themes and subthemes are presented in the order of highest frequency of occurrence, which is defined by the number of times the theme was cited by the participants in their interviews. Quotations from the interviews are also provided.

### Research objective 1: How has the COVID-19 pandemic impacted the mental health of families with children who have ADHD?

Table 2 provides a summary of the ways in which the COVID-19 pandemic affected the mental health of families with children who have ADHD. The most frequently identified mental health effects were increased child anxiety, social disconnectedness, and negative impacts on parental mental health.

**Theme 1: Increase in child anxiety (frequency 53).**   Parents expressed that their children experienced increased anxiety during the COVID-19 pandemic. The four prevalent subthemes were attachment issues, difficulty navigating online learning from home, lack of structure and routine, and fear of COVID-19. One parent noted:

> He is more anxious, highly anxious. He has a lot more of symptoms of anxiety, super anxious all the time [. . .] He's afraid about a lot of things. He's very confused about the pandemic, like one hundred and fifty percent confused. (Participant 6, parents, female)

*Subtheme 1a*: *Fear of COVID-19 (frequency 15)*. Parents described a fear of COVID-19 as a contributor to increased child anxiety. This subtheme included factors such as children fearing contracting COVID-19 themselves, as well as loved ones contracting COVID-19. The following excerpts illustrate the fear experienced by children:

**Table 2. Frequency summary of main themes and subthemes of how COVID-19 affected mental health of families with children who have ADHD.**

| Theme | Frequency |
|---|---|
| Increased Child Anxiety | 53 |
| • Fear of COVID-19 | 15 |
| • Attachment | 9 |
| • Difficulty Navigating Online Learning | 6 |
| • Lack of Structure and Routine | 4 |
| Disconnectedness | 41 |
| • Social Isolation | 27 |
| Deteriorating Parental Mental Health | 35 |
| • Lack of Social Support | 11 |
| • Difficulty Managing Parenting Duties | 7 |
| • Parent-Led Schooling | 7 |
| • Increased Parental Anxiety | 4 |
| • Concern for Health of Family Members | 3 |
| Increased Child Depression | 13 |
| • Grief | 2 |
| • Lack of Goals | 1 |
| • Concern | 1 |
| Increased Calm | 7 |
| • Family Time | 4 |
| • Reduced Stress | 3 |
| Uncertainty | 6 |
| • School and Online Learning | 1 |
| • Future | 1 |
| Improved Child Mood | 5 |
| Decrease in Child Self-Efficacy | 1 |

> There is a kid in school who's relative got the virus [. . .] It's sad to know that lots of people are just dying because of the sickness, which today normally you wouldn't think that would really hurt someone or make them pass away, but it's something that happens every day for lots of people now. (Participant 8, child, male)

> He has gone through some phases of increased anxiety, especially at the beginning of the pandemic. [It] took a long time to understand what was behind that. I work in health care and was redeployed to an area of working with people where the exposure would be a higher risk. And I didn't realize that he was even aware of what the implications were until one day he burst into tears and said that he [wouldn't] like living in an orphanage. And I was like, wow. (Participant 11, parent, female)

*Subtheme 1b*: *Attachment (frequency 9)*. Parents described their children's attachment being negatively affected during the pandemic in a way that contributed to increased child anxiety. The subthemes of attachment included factors such as spending too much time together, separation anxiety, and trouble sleeping alone. One parent noted:

> He's not able anymore to sleep by himself in his own room [. . .] His anxiety level has definitely spiked during COVID. His attachment issues are a result of it just being the two of us and spending so much time together. (Participant 1, parent, female)

*Subtheme 1c*: *Difficulty navigating online learning from home (frequency 6)*. Children participants reported experiencing increased anxiety due to the difficulty of navigating online learning from home. Both parent and children participants described online learning as a challenge

in their interviews. This theme included issues with technology, lack of one-on-one teacher support, parents juggling both work and helping with online school, and insufficient instruction for operating the virtual platform. Additionally, parents often cited children's ADHD as an additional barrier to online learning, further contributing to increased child anxiety. Below is an example that illustrated the difficulty of navigating learning from home.

> [Online learning] came at a hard time where she really needed us. It's not just the ADHD. It's all the other stuff. It's a very volatile situation. Really stressful because now suddenly, your parents are teaching you and all of your shortcomings are becoming very evident. And so, the stress, right? That was a really difficult thing to do. (Participant 3, parent, female)

*Subtheme 1d*: *Lack of structure and routine (frequency 4)*. Parents described a lack of structure and routine for their children as a contributing factor to increased child anxiety, especially as routine is essential for managing many ADHD symptoms [58–60]. This subtheme included factors such as cancelled activities, school closures, and loss of past routines. Below is an example that illustrates the challenges associated with a lack of structure and routine.

> [x] is very routine oriented and structure oriented. When school closed for a couple of weeks, we could deal with that. But then when they didn't reopen, and I work full time, and we lost all sense of structure, and we were not able to engage with online school, I just could not figure out how to make that work. And things just fell apart at that point. (Participant 11, parent, female)

**Theme 2: Disconnectedness (frequency 41).**   Participants described feeling disconnected from family, friends, and regular life due to the COVID-19 pandemic and emphasized its effects on mental health. Within the theme of feeling disconnected, the subtheme of social isolation was identified.

*Subtheme 2a*: *Social isolation (frequency 27)*. Participants described social isolation as contributing to feeling disconnected during the COVID-19 pandemic. Participants described feeling lonely due to social isolation guidelines. One parent noted, "We've had lots of tears. We've had serious crying sessions, serious concerns about and expressions of I'm not doing well. Things like, I'm sad. I feel scared. I feel alone. I don't have anyone to hang out with." (Participant 14, parent, female)

**Theme 3: Negative impact on parental mental health (frequency 35).**   Parents described experiencing a negative impact on their mental health due to the COVID-19 pandemic. The subthemes included increased frustration and stress around parent-led schooling, difficulty managing increased parental duties, lack of social support, increased anxiety, and concern for the health of family members.

*Subtheme 3a*: *Lack of social support (frequency 11)*. Parents described a lack of social support throughout the COVID-19 pandemic contributing negatively to their mental health. This subtheme included aspects such as a lack of childcare options, lack of social interaction and lack of physical contact. One parent noted:

> It's been really hard, all of the social support that we had, all of the people who gave us a break and gave our kids a break, are not there. [My children's] friends that they used to hang out with and have sleepovers with and socialize with and mutually support, that's not available either. So, as parents, as a family, and also the kids as individuals, are now down to

zero social support and no physical contact with the world. And that's pretty tough. (Participant 14, parent, female)

*Subtheme 3b*: *Difficulty managing parental duties (frequency 7).* Participants described having trouble managing parental duties during the COVID-19 pandemic, resulting in deteriorating parental mental health. This subtheme included factors such as working from home while caring for their children, and feeling guilty about struggling with parental duties. Several parents noted:

Before COVID-19 we were able to go to places. Like he mentioned, he loves to be at the library. There were always some limitations of being at home. But since COVID-19 began, it has been the most painful thing, I cannot even describe [it]. As I said, it is exhausting. It's been the most severe, painful thing I have ever gone through. It has been challenging caring for him medically since he was born, but the most challenging time has been during COVID-19. I feel like I have been defeated and I don't know what else to do. I'm defeated, actually, as I speak, I'm defeated. (Participant 6, parent, female)

[My husband and I] work in a similar field. So, we're both doing counselling with individuals from our kitchen and from our dining room while we're trying to manage two kids who were being homeschooled. I never signed on to homeschool because that's not my strength. And I never signed on to be a stay-at-home parent because that's really not my ideal. I do love my kids and they know that. But this has been hugely negative for a person like me. I changed jobs because of this. (Participant 14, parent, female)

*Subtheme 3c. Increased frustration and stress surrounding parent-led schooling (frequency 7).* Participants described increased frustration and stress due to parent-led schooling and lack of information regarding school guidelines. This subtheme included factors such as parents struggling to help their children with virtual learning, technology frustrations, and not being informed about school guidelines during the pandemic. One parent noted "They need to go back to school. That's for my mental health. (Participant 5, parent, male). Another parent noted:

I needed to hire a tutor to help support things more and give me a break. [My son] does not want me as his teacher. I already fill so many roles in his life, he doesn't want me in that one. It causes a lot of stress and frustration for both of us. (Participant 2, parent, female)

*Subtheme 3d. Increased parental anxiety.* Parents described an increase in their anxiety because of the COVID-19 pandemic. This subtheme included factors such as fearing the unknown aspects of COVID-19 and anxiety surrounding parenting. Two parents noted:

I just wanted to stay away from people. I felt like in our home we had at least this nice little cocoon. We were safe. And then if anybody had to step out, there was just so much unknown, and I cried. I remember the first time I was about to go to Costco, I cried. I said to my husband, I don't want to go. I had the membership. That was the problem. I [was the one] that had to go. (Participant 9, parent, female)

My son's screentime has doubled, which makes me, as a parent, feel guilty and anxious [. . .] It's gut wrenching. But when you're at home and you're trying to work, what else can you do? But you still feel awful about it. (Participant 12, parent, female)

## Research objective 2: How has the COVID-19 pandemic created barriers to maintaining optimal mental health for families with children who have ADHD?

Table 3 provides a summary of the parent-identified barriers to positive mental health for their families during the COVID-19 pandemic. The most frequently identified barriers were lack of routine, lack of social interaction and social support, and uncertainty.

**Theme 1: Lack of routine (frequency 11).** Parents described the lack of routine as the most significant barrier to positive mental health for themselves and their children with ADHD. This theme included factors such as a lack of extra-curricular activities, minimal school involvement, and being stuck at home. One parent noted:

> [My son] is tired of the routines just recycling in the house. He [can't] go and express his brain somewhere else. I don't know how long this is going to last and I don't know how we're going to survive. (Participant 6, parent, female)

**Theme 2: Lack of social interaction and social supports (frequency 10).** Participants described the lack of social interaction during the pandemic as the second most salient barrier to maintaining optimal mental health. This theme included factors such as not being able to see family, friends, or coworkers. Two parents noted:

> [The biggest barrier. . .] that's easy, social interaction! Because we can't see our coworkers and other extended family members, when we go for walks, we're more likely to stop and say hello to a neighbour and make friends with other kids down the road, and things like that. So, in the last six months, we've transitioned our social interactions in a way. While these [new] interactions are lovely, they are not the same as a warm body sitting beside you and having a cup of tea. So, [while] we have gotten to know our neighbours a little bit better by discussing our different stresses. . . they're not really digging deep. They're very superficial conversations, but it's fulfilling something that we obviously need. (Participant 9, parents, female)

> I [ask myself], have I left the house today? Have I interacted with anybody besides my eight-year-old? And I think that the hardest part for me has been the lack of adult exposure. I don't remember the last time I just went out for lunch with a friend or said 'come over tonight and have a glass of wine.' (Participant 11, parent, female)

**Table 3. Frequency summary of main themes and subthemes of barriers to positive mental health for families during the COVID-19 pandemic.**

| Theme | Frequency |
|---|---|
| Lack of Routine | 11 |
| Lack of Social Interaction and Social Supports | 10 |
| Uncertainty and Fear | 8 |
| • School | 3 |
| • Fear of COVID-19 | 2 |
| • Childcare | 2 |
| Lack of Access to Services | 3 |
| Lack of Alone Time | 3 |
| Lack of Physical Activity | 2 |
| Disruption and Change | 1 |
| Removal of Positive Behaviour Reinforcements | 1 |

**Theme 3: Uncertainty and fear (frequency 8).** Participants described uncertainty and fear as a barrier to positive mental health. Within uncertainty and fear, the subthemes identified were uncertainty about childcare, fear of COVID-19, and uncertainty about school.

*Subtheme 3a*: *Childcare (frequency 3)*. Parents described the uncertainty surrounding childcare as a barrier to positive mental health. This theme included aspects such as parents not knowing how to balance additional childcare responsibilities while children stayed home while simultaneously working, and uncertainty surrounding finding childcare during social isolation mandates. One parent noted, "The constant uncertainty of childcare [was the biggest barrier]. And what were we going to do? How was this going to work? You can do anything for a short period of time, but it's not sustainable." (Participant 11, parent, female)

*Subtheme 3b. Fear of COVID-19 (frequency 2)*. Participants described the fear of COVID-19 as a contributing factor to their uncertainty and fear throughout the pandemic. This subtheme included factors such as a general fear of COVID-19 due to the lack of knowledge about the virus, parents fearing contracting COVID-19, and parents worrying that their children or loved ones would contract COVID-19.

> With schools opening, a few parents are [just sending their kids and hoping for the best]. But with my kid, I can't [do that]. First, I'm really worried [in general about the virus]. And if [my son] is to get sick, it will be a big alarm in my house. So, it's been the most stressful thing. I [recently] had a new baby in the middle of the COVID-19 pandemic too. And with [my son who has ADHD], it's like having 10 babies, plus the new baby. (Participant 6, parent, female)

*Subtheme 3c*: *School (Frequency 2)*. Participants described the uncertainty surrounding school as being a barrier to positive mental health. This theme included factors such as uncertainty and frustration surrounding the use of the virtual learning platform, and uncertainty surrounding school guidelines, such as when school would return to in-person, and the rules in place once it did. One parent noted:

> I think for [my child], it was the sudden disruption in routine and the uncertainty. First, it was for two weeks, then it was for seven weeks. Then it was like, we don't know if schools are going to open again. [There was] prolonged uncertainty and not knowing was really, really hard. You can do anything for a short period of time, but it sustained. By July [2020] we were so done with the way things were. And then the entire uncertainty about school returning [in the fall] and all of August right up until the week before school started, [we didn't know] if school was going to happen. And so that I think, for him, was very difficult because he thrives on understanding what's going on and he asks a lot of questions. And there were no good answers. There was nothing I could say because nobody could tell us what the plan was. (Participant 11, parent, female)

## Discussion

The current study examined how the COVID-19 pandemic impacted the mental health of families with children who have ADHD. The most frequently identified mental health effects on families were increased child anxiety, feelings of social disconnectedness, and deteriorating parental mental health. The most frequently identified barriers to maintaining optimal mental health were lack of routine, lack of social interaction and social supports, and uncertainty and fear. The following will unpack the major themes and offer suggestions for future support programs.

### Research questions 1: How has the COVID-19 pandemic impacted the mental health of families with children who have ADHD?

**Increase in child anxiety.**   Parents described increased child anxiety as the most salient effect of the pandemic on their mental health. First, parents described this increase in anxiety manifesting in their children as an overwhelming fear of COVID-19. Both parent and child participants said this fear included being afraid to get sick, being afraid that loved ones would get sick, and general fear surrounding the unknowingness of the virus. The fear of the unknown surrounding COVID-19 has been shown to predict increased mental health issues among adults [1–3,19], and clearly has similar implications for children with ADHD as well.

Second, parents described the increase in their children's anxiety manifesting as a "clingy" attachment style due to the stay-at-home orders and general social isolation. Many described that their children became significantly less independent, were no longer able to sleep alone, and would display more anxiety during separation than prior to the pandemic. This created long-term issues when in-person school temporarily resumed as children struggled to leave their parents and experienced heightened anxiety. Without a looming pandemic, children with ADHD are already at a 15% to 50% higher risk than typically developing children to have an anxiety disorder, including social anxiety, separation anxiety and generalized anxiety [11,76]. Among children with ADHD who have comorbid separation anxiety, more than 40% continue to experience separation anxiety as adolescents [77] and adults [78]. The current findings underscore the importance of identifying both short-term and long-term supports for children and families to help mitigate the cascading effects of childhood anxiety into adulthood.

Third, parents frequently related their children's increased anxiety to the difficulties navigating online learning. Schools and teachers had to make drastic and hurried changes to their instructional approaches according to rapidly evolving ministry of education guidelines. This created a breeding ground for technological issues, student and parent confusion, and insufficient one-on-one supports for children with exceptionalities. Children with ADHD are already more likely to experience academic challenges compared to their typically developing peers [79–81], and the suboptimal learning conditions unsurprisingly further increased children's anxiety around schooling.

Fourth, both parents and children emphasized the change in daily routine and consequent lack of structure as a contributor to increased child anxiety. Regarding schooling, the shift from in-person schooling to online schooling interfered with morning routines, eliminated recess, peer interactions, transitions between classes, and left gaps of unstructured time. Children with ADHD thrive on predictable routines to help mitigate inattentive and hyperactive symptoms [82,83], as well as to help minimize externalizing (e.g., conduct issues) and internalizing (anxiety) behaviours [83]. Parents described school hours as being inconsistent, their own work hours being inconsistent, and having to cancel extracurricular activities that were vital outlets for their children's energy. While previous findings with typically developing children have also shown that the pandemic has led to increased anxiety [7,84], results from this study point to likely differences in the magnitude of negative experiences between these groups and potential differences in the long-term effects.

**Feeling disconnected.**   Both parents and children expressed feeling disconnected from friends and family due to social isolation mandates. Previous work has shown that social isolation can be detrimental to both physical and mental wellbeing, and that the effects can be long-lasting [68]. This is especially problematic for children as social isolation in childhood and early adolescence can result in immediate and long-term deficiencies in connectedness, feelings of depression, anxiety, and somatic symptoms [85–90]. Loneliness due to social

isolation in childhood has also been shown to predict higher levels of depressive symptoms and other age-related diseases in the stress-sensitive biological systems later in life [87,90]. Furthermore, social isolation in childhood can result in poor sleep patterns, sleep difficulties, and decreased executive functioning, including the use of self-control and time management [82,85,91]. These are all symptoms that children with ADHD already commonly experience and may become exacerbated during social isolation leading to compounding negative outcomes. Indeed, many parents noted that their children's sleep difficulties increased during the pandemic, and that their ADHD symptomology worsened [12].

**Deteriorating parental mental health.** The current findings emphasize the profound effects of the pandemic on diminishing parental mental health. Importantly, parents of children with ADHD tend to have a significantly higher baseline of stress than parents of typically developing children [92] as the demands of caring for a child with ADHD are often much greater [92,93]. Parental stress has been shown to negatively impact all children [31,32] but especially those with ADHD [92]. Parental stress is associated with less structured child routines and maladaptive parenting behaviours such as harsher and less consistent discipline. Critically, the removal of structured routines erodes an effective strategy for managing ADHD symptomology [82,83,94], and maladaptive parent-child interactions are related to increased child misconduct and anxiety, which are highly comorbid with ADHD [92,95–97].

The first largest contributor to worsening parental mental health was the lack of support. Many parents expressed having strong support systems prior to the pandemic. For example, parents had nannies, babysitters, friends, and family who helped with childcare and provided social interaction. Support systems are critical for helping reduce psychological distress among caregivers of children with ADHD [98,99]. Parents also described having professional support for their children such as mental health professionals and programs targeting ADHD management that were no longer available. Other research during the pandemic has similarly shown that parents with children who have special needs lost access to informal and formal respite, which they relied on to decrease stress, reduce isolation, and experience short-term relief [100].

The second contributor to declining parental mental health was the difficulty managing parenting duties given the lack of social support and increased caretaking demands. Parents expressed struggling to juggle the diversity of roles thrust upon them including teacher, full-time caregiver, home keeper, and employee. Many parents described relying on their children being in school in order to work and offload childcare. Whether parents worked from home or worked outside the home, they faced numerous childcare issues. Many parents expressed feeling tremendously guilty about struggling with parental duties and not doing enough for their children, which contributed to their declining mental health. Given that parents with children who have ADHD already struggle with the burden of care, feelings of guilt and shame, and managing occupational responsibilities [13–18,101], the pandemic compounded these issues and led to further deteriorating mental health. As some parents expressed, the cascading effects often included worse parent-child interactions and further impairment in ADHD symptomology.

The third contributor to declining parental mental health was parent-led schooling. Parents expressed frustration having to take on the role of teacher with poorly defined guidelines; they did not know for how long school would be virtual and were provided few instructions on how to navigate the curriculum and the online platform. Parents also expressed not having the time to help their children due to their own jobs and resulting feelings of guilt and inadequacy. Furthermore, parents described feeling ill-equipped to support their child's unique ADHD needs in a learning environment, leading to intensifying feelings of frustration and guilt. The challenges with online learning environments for parents, teachers and students of various

ages is well supported [102–104]. However, minimal work has examined the unique impact that parent-led schooling has had on children with exceptionalities and their parents. The current study emphasizes the need for continued individualized and directed instruction for children with ADHD and other exceptionalities, as many parents are not trained to effectively support these children's learning needs.

The fourth contributor to diminishing parental mental health was increased parental anxiety. Parents cited several sources for their increased anxiety such as fear of COVID-19, feeling overwhelmed with household, parental and workplace duties, and feelings of inadequacy when it came to parenting specifically. This is especially problematic given that parents of children with ADHD typically have a higher incidence of anxiety, depression and substance abuse compared to parents of typically developing children [105,106]. As previously mentioned, the cascading effects of increasing parental anxiety are often poorer parent-child interactions [31,32] and correspondingly worsening ADHD symptomology [12]. This vicious cycle can create an overwhelming and helpless environment for all family members. The section on future directions will offer suggestions on how to better support families with children who have exceptionalities during times of increased familial stress and crisis.

### Research question 2: How has the COVID-19 pandemic created barriers to maintaining optimal mental health for families with children who have ADHD?

**Lack of routine.**   Parents indicated that lack of routine during the pandemic was the most salient barrier to maintaining optimal mental health. As previously mentioned, the pandemic dramatically altered family routine, which has been identified as a contributor to poor mental health for all families during the pandemic [36,107]. However, the maintenance of routine is especially important for families who have children with ADHD as predictable daily structure is an effective way to manage ADHD symptoms [82,83,94]. Many parents noted that the removal of their children's extracurricular activities was a major detriment to routine, social interaction, and ability to release energy. These findings underscore how mental health supports should focus on adding structure and routine to children's and families' lives. This could be implemented through more structured and better-executed virtual schooling, by offering online avenues for extracurricular activities, and offering more regularly scheduled virtual support programs for families so that there is a consistent time and place in their schedule for interaction.

**Lack of social interaction and social supports.**   Both parents and children frequently identified the lack of social interaction and social supports as barriers to maintaining optimal mental health. Although participants indicated that they were able to interact with family and friends virtually, they expressed that this was insufficient. Many also indicated experiencing "zoom fatigue" and burnout [108] from virtual communication as this was the primary method of communication for work and school. Parents also indicated losing their social supports including childcare and mental health services for themselves and their children. The loss of social support for any family can have devasting consequences, but even more so for families with children who have ADHD as they also lose vital mental health supports that help mitigate higher baseline levels of parental stress [92,93]. Greater efforts need to be geared towards providing virtual programming that promotes social connection and interaction, especially for struggling families. Further guidelines and allowances for interacting in-person in a safe manner (e.g., outdoors, distanced, masked) should also be developed to help families feel connected and supported during times of crisis.

**Uncertainty and fear.**   The uncertainty surrounding school, fear of the COVID-19 virus, and the uncertainty surrounding childcare were notable barriers to mental health. With

respect to school, many parents and children expressed feeling confused and unsupported throughout the virtual school experience. Parents were also unsure when their children would return to in-person schooling and could not communicate helpful timelines to their children. When given the option to return their children to school, parents expressed receiving insufficient guidelines as to how in-person schooling would proceed and the safety precautions instituted to contain any spread of infection. Whether parents opted into the continued uncertainty of navigating an online learning environment, or whether parents sent their children to an ill-defined school setting, they all expressed experiencing psychological distress due to such uncertainties.

Participants also expressed fear of the COVID-19 virus as being a significant barrier to their mental health. Due to the dearth of knowledge surrounding the virus at the beginning of the pandemic, participants were uncertain about proper safety measures and were fearful of their families becoming sick and dying. It should be acknowledged that the COVID-19 pandemic was unprecedented and thus information was scare at its onset. However, providing families with mental health supports to help them cope with increased anxieties and fears of the unknown is imperative to mitigate the compounding effects of familial stress and worsening ADHD symptomology [2,3,19].

The uncertainty surrounding childcare was also a notable barrier to mental health. Parents with children who have ADHD already experience increased childcare demands [13–18,101]. These parents were also required take on the additional roles of teacher and full-time caregiver. They expressed feeling uncertain about their new roles, especially as teacher, as they were not trained to provide the kind of instruction children with ADHD require to thrive. Additionally, many parents who worked outside the home did not know how to find childcare due to schools, babysitters, nannies, family members, and other childcare services being unavailable. Parents expressed feeling elevated levels of guilt surrounding their lack of availability to support their children and provide them with proper childcare. Children similarly expressed frustration around the ill-defined childcare dynamics. Uncertainty and fear increase parental stress and child anxiety [109–111], exacerbating ADHD symptomology and once again feeding into the vicious cycle of familial stress and worsening mental wellbeing.

The current study underscores a clear disruption to several component elements of the family system. The second most significant theme identified in the data across both research objectives was disconnectedness and social isolation. This reflects a salient fracture among the complex and ecological systems as they rely on an interconnectedness among the family unit and the broader community including school, friends, peers, places of employment, therapy groups, and many other parts, in order to function successfully [52–54]. Children's increased anxiety levels, parents' declining mental health, and the constant overtone of uncertainty and fear also underscores the inability of families to return to a homeostatic balance, likely related to disruptions in other component elements as a function of social disconnection. As previous research has strongly noted, families dealing with ADHD rely heavily on structured routine to remain well-functioning [82,83,112]. The lack of routine imposed by COVID-19 was noted as the most salient barrier to maintaining mental wellbeing among the families in the current study. This highlights how certain practices are essential to maintaining equilibrium among families with ADHD, and once chronically disrupted, can lead to the mental health struggles in the family system (e.g., increased child anxiety, deteriorating parental mental wellbeing). This is the first study to directly apply Systems Theory to understanding the impact of COVID-19 on families dealing with ADHD. This theoretical approach has the potential to offer a rich understanding of how critical elements of diverse family systems are impacted by environmental pressures, and why greater dysfunction may exist within certain dynamics, with the ultimate goal of offering supports for those who may need it the most.

Although the current study did not formally evaluate the attachment styles of the children or parents who participated, the findings align with the notion that families dealing with ADHD may have experienced exacerbated struggles during the pandemic due to potential disharmony in child-parent attachment. Children with ADHD experienced greater anxiety, parents experienced a deterioration in their mental health, and there were debilitating levels of uncertainty and fear in the family system. Prior Attachment Theory work involving neurotypical individuals similarly found that children and parents with insecure attachment experienced similar declines in their mental wellbeing during the pandemic [60,66], but that individuals with secure attachment were more resilient against the pandemic's psycho-emotional impacts [113]. Given that children with ADHD and their parents often exhibit more insecure attachment [62–65], it is unsurprising that the families in the current study experienced exacerbated hardships. While we cannot confirm that the children in our study or their parents were insecurely attached, the evidence points to greater familial challenges managing the additional stressors brought on by the pandemic. It would be beneficial for additional research to directly assess attachment styles among children and parents in a family dealing with ADHD and to relate them to their experiences during the pandemic. This would offer novel insight and applicability of Attachment Theory to pandemic-related or environmental-stress research among families dealing with ADHD.

Taken all together, the current study provides critical insight into how families with children who have ADHD experienced the pandemic and the notable effects on their mental health. Families worldwide experienced similar mental health effects [2,3,24], but the magnitude of these effects is greater for vulnerable families. The themes found in this work can help inform support programs for families with children who have ADHD and other exceptionalities in times of crisis and social isolation.

## Future directions: Support programs

One of the most important support programs for families with children who have ADHD is one-on-one educational support. Children with ADHD require targeted support during their learning to thrive. School boards and universities could utilize their network of teachers and faculty members, (including part-time teachers, substitute teachers, support staff) to provide greater learning support. Tutors could be matched with children on a need's basis, providing children with personalized support and offloading the burden of educational care from parental shoulders. Although several institutions took it upon themselves to mobilize and generate such initiatives, they were small scale and only affected specific cities. With proper governmental support and funding, such initiatives could be scaled up for broader and more effective reach.

Another critical support program would involve providing extracurricular outlets for children with ADHD that take into consideration their specific needs, as healthy outlets for their energy are directly related to their mental wellbeing. Schools or community centers could provide more online or socially distanced extracurricular opportunities for children that follow a strict schedule. This would not only allow children with greater access to safe activities, but it would also help infuse structure and routine into their turbulent circumstances. Extracurricular programs would also help facilitate social interaction and minimize feelings of disconnect.

Additional support programs for both parents and children should include avenues for social interaction and connection. For children during times of online schooling, this could easily be facilitated by scheduling time in the school day for students to chat with one another through virtual breakout rooms. As shown in the current study, many children with ADHD experienced strong emotions during the pandemic, including social isolation and loneliness.

Having the time and space to discuss these feelings or simply to socialize with peers could help these children process their emotions while also providing more opportunity for social interaction. Community centers and libraries could also offer social programming for families and children with exceptionalities to enable a sense of connection and foster some routine and extracurricular engagement. For parents, it is evident that social support is critical, whether that is informal (family and friends) or formal (support groups, individualized counselling), as feeling connected to others may buffer against the negative psychological impact of the COVID-19 pandemic. It may also be beneficial for the Ministry of Health to recognize the impact of crises on families with children who have special needs and create programs that connect individuals with similar lived experiences to offset the negative psychological consequences that inevitably coincide with times of crisis.

Families should also be made more aware of existing online and self-help training programs that are designed to help parents develop communication and engagement skills with their children who have ADHD [114–116]; this is especially important as parent-child interactions often worsen during times of high-stress [12,31,32]. Previous work has also shown that offering families dealing with ADHD telephone-assisted self-help programs reduces negative parenting behaviour, and promotes adherence to ADHD medication [117]. A potential option would be for telehealth services to form a distinct unit and offer targeted support for families that have children with ADHD and related disabilities. These services can offer support for children surrounding ADHD-specific mental health concerns, as well as education for parents about ADHD and related mental health issues. Similarly, existing parenting programs should seek to explore whether modifications could be made to be more appropriate for delivery during pandemic-related restrictions so that continued resources and routine are offered to families dealing with ADHD [118].

## Limitations

While this research contributes to our understanding of the effects of the pandemic on families with children who have ADHD, it is not without its limitations. The first limitation to consider is the sample size. Other families may have experienced the pandemic differently than those that participated in the current study, and thus may have encountered different effects on mental health or barriers to maintaining optimal mental health. Although a larger sample size may have captured more diverse insights, researchers noted a large degree of consistency in the perspectives of families interviewed. The second limitation to consider is the use of virtual interviews as this may have limited participation of families who did not have access to the internet or a computer. And third, the themes identified in the study predominantly reflected parent perspectives as it was challenging for children to discuss their experiences with the pandemic. The questions asked in parent interviews were more expansive and thorough whereas the questions asked in child interviews were simpler and therefore unveiled less information. As a result, this may have limited the study's ability to fully reflect how children with ADHD experienced the pandemic.

## Conclusion

The current study examined how the COVID-19 pandemic impacted the mental health of families with children who have ADHD. The most frequently identified mental health effects on families were increased child anxiety, feelings of social disconnectedness, and deteriorating parental mental health. The most frequently identified barriers to maintaining optimal mental health were lack of routine, lack of social interaction and social supports, and uncertainty and fear. It is critical for future work to conduct follow-up studies with families to assess the long-

term effects of the pandemic on their mental health and to identify ways to offer continued support. Overall, this work brings voice to the families of children with ADHD and contributes to our understanding of the pandemic's impact on the wellbeing of vulnerable families. We hope the findings will contribute to the creation of safeguards for mental wellbeing in times of crisis.

## Supporting information

**S1 Appendix. Parent interview guide.**
(DOCX)

## Acknowledgments

The authors would like to thank all families who participated in this study. Their vulnerability to discuss their circumstances is invaluable in the pursuit to better support families during times of crisis.

## Author Contributions

**Conceptualization:** Alexis Winfield, Barbara Fenesi.

**Data curation:** Alexis Winfield, Carly Sugar, Barbara Fenesi.

**Formal analysis:** Alexis Winfield, Carly Sugar, Barbara Fenesi.

**Investigation:** Alexis Winfield, Carly Sugar.

**Methodology:** Alexis Winfield, Barbara Fenesi.

**Project administration:** Barbara Fenesi.

**Resources:** Barbara Fenesi.

**Software:** Alexis Winfield, Barbara Fenesi.

**Supervision:** Alexis Winfield, Barbara Fenesi.

**Validation:** Carly Sugar, Barbara Fenesi.

**Visualization:** Carly Sugar.

**Writing – original draft:** Carly Sugar, Barbara Fenesi.

**Writing – review & editing:** Alexis Winfield, Carly Sugar, Barbara Fenesi.

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
