## [Decision Letter · Decision Letter 0]

15 Sep 2022

PONE-D-22-20962The Impact of the COVID-19 Pandemic on the Mental Health of Families Dealing with Attention-Deficit Hyperactivity DisorderPLOS ONE

Dear Dr. Fenesi,

Thank you for submitting your manuscript to PLOS ONE. After careful consideration, we feel that it has merit but does not fully meet PLOS ONE’s publication criteria as it currently stands. Therefore, we invite you to submit a revised version of the manuscript that addresses the points raised during the review process.

We look forward to receiving your revised manuscript.

Kind regards,

Tai Ming Wut

Academic Editor

PLOS ONE

Journal Requirements:

Reviewers' comments:

Reviewer's Responses to Questions

**Comments to the Author**

1. Is the manuscript technically sound, and do the data support the conclusions?

Reviewer #1: Partly

Reviewer #2: Yes

2. Has the statistical analysis been performed appropriately and rigorously? 

Reviewer #1: No

Reviewer #2: Yes

3. Have the authors made all data underlying the findings in their manuscript fully available?

Reviewer #1: Yes

Reviewer #2: Yes

4. Is the manuscript presented in an intelligible fashion and written in standard English?

Reviewer #1: Yes

Reviewer #2: Yes

5. Review Comments to the Author

Reviewer #1: Overall impression

I understand that interview studies are usually exploratory in nature. However, the findings from this study (e.g., Increased child anxiety and disconnectedness, deteriorating parental mental health, barriers to maintaining optimal mental wellbeing due to the lack of routine, the lack of social interaction, and the lack of social support) were not surprising nor gave the field significant insights. The results were not intellectually stimulating because all parents around the world have been suffering from the same things in the past 3 years. This has made the results of this study less significant to the ADHD population specifically.

Given that a lot of countries start to stop implementing social distancing policies, the findings from this study might not be timely enough. The findings might still give the field some insights if the next pandemic has come to us, however, the authors have only mentioned some general suggestions to help ADHD children. For example, providing greater learning support, providing online extra-curricular activities as energy outlets, and enhancing social interactions. These suggestions are true and important for all students, not specifically for ADHD children, making the theme of this paper (i.e., ADHD caregiving) less coherent. Sharing regarding proactive measures to face another pandemic for ADHD kids specifically could be added to the discussion to deepen the discussions.

Research rationale and theoretical foundation

Again, I understand that interview studies are exploratory in nature. However, the theoretical foundation driving this research study was too weak. The academic research gap has not been clearly justified or identified either. The introduction was a bit too short, and the literature review in the introduction was too sketchy and descriptive, which might be one of the reasons the authors were not able to derive the research motivation clearly. Without a clear theoretical framework, the academic quality of an article might be threatened. In the introduction, the authors have mentioned the impact of the COVID-19 pandemic and how families have been affected. They have also provided some reasons why they were particularly interested in the ADHD population. However, to me, they were not academically convincing reasons or motivations to conduct this research. Academic quality could be enhanced by theoretical frameworks. The discussion and review of extant literature under a certain theoretical background might help researchers think of real and academically sound reasons to conduct academic research.

Methodologies and Results

The sampling rationale was unclear. The reason why ADHD kids must be between 7 and 12 years was not clear. The sentence “Living with them for at least some of the time” was ambiguous. The meaning of “at least some of the time” was unclear. Without a clearly written method section, the replicability of this study will be seriously affected.

I am not sure about the objectives for interviewing two groups of participants (i.e., parents and their ADHD children) in this paper. It makes more sense to me that the authors wanted to compare these two groups of participants on the same topic, so they interviewed them separately. For example, hypothetically, a group of researchers wanted to study the well-being changes of parents of ADHD kids and ASD kids during the COVID-19 pandemic. Therefore, they recruited 30 parents of ADHD kids and ASD kids respectively, and they were asked the same set of questions. By doing the content analysis, a different set of themes might be found in these two categories of parents. However, this study did not do a similar comparison, parents and kids were asked different questions. This made me confused about the reason why that was.

Table 2 Page 10: the sub-themes might be a bit redundant. For example, the lack of social support may be a reason why parents thought it was difficult to manage parenting duties. Could they be grouped together? The increased parental anxiety might not be a sub-theme, as it was a manifestation of the main theme “deteriorating parental mental health”.

Other minor issues:

Page 3 line 66 (referencing style is not consistent).

In some sections, the writing styles might be a bit too casual. For example, between page 22 and page 24, when the authors discussed the contributors to declining parental mental health, the authors repeatedly wrote “many parents” (e.g., “many parents had strong support system prior to the pandemic”), would it be better to change to sentences with milder tone “a strong support system might be more available for parents prior to the pandemic”?

Thank you very much for your hard work.

Reviewer #2: The authors have:

1. Chosen an important topic that deserves timely research, discussion, and follow-up;

2. Demonstrated sufficient analysis and discussion of previous studies in the literature review;

5. Conducted thorough analyses; &

6. Provided the way forward.

Minor Revisions are required. Please address the following comments and questions.

1. Consider converting this paragraph (line 118) to an objectives section.

2. Include a separate research gap section and identify the research gaps;

3. Add response rate;

4. Any theory guiding the design of interview questions? Were there enough questions?

5. Sample size issue: Were attempts made to interview more participants?

6. PLOS authors have the option to publish the peer review history of their article (what does this mean?). If published, this will include your full peer review and any attached files.

Reviewer #1: No

Reviewer #2: No

---

## [Author Response · Author response to Decision Letter 0]

12 Oct 2022

Academic Editor

1. Please ensure that your manuscript meets PLOS ONE's style requirements, including those for file naming

Response: We have formatted the manuscript according to PLOS ONE’s style requirements. 

2. We note that you have indicated that data from this study are available upon request. PLOS only allows data to be available upon request if there are legal or ethical restrictions on sharing data publicly. 

Response: We have indicated our response in the revised cover letter. The response is as follows:

There are ethical considerations surrounding sharing of interview-based data. Given the qualitative interview-based nature of the data, there is personal, detailed, and sensitive information contained within the transcripts that does not lend itself to ubiquitous public access. Although the interview data can be de-identified, they cannot be redacted in a way that makes the content completely de-identifiable (e.g., the sharing of personal experiences and stories could potentially be recognized by someone). Redaction could also potentially remove vital data for replication. Western University’s Research Ethics Board has imposed data sharing restrictions as they do not allow identifiable or de-identified data to be electronically transmitted outside of Western including open access repositories. Requests can be sent to Western University’s Office of Human Research Ethics (Phone: 519.661.3036; Fax: 519.850.2466; email: wrem@uwo.ca). Requests can also be sent to the corresponding author (bfenesi@uwo.ca).

3. PLOS requires an ORCID iD for the corresponding author in Editorial Manager on papers submitted after December 6th, 2016. Please ensure that you have an ORCID iD and that it is validated in Editorial Manager.

Response: The corresponding author’s ORCID iD has been updated.

Response: We have included an ethics statement on page 11 (line 242-243) in the Methods section of the manuscript. We have also indicated that written consent and assent were provided by parents and children, respectively on page 8 (line 173). 

5. Please include captions for your Supporting Information files at the end of your manuscript, and update any in-text citations to match accordingly.

Response: We have included captions for the Supporting Information file and labelled it appropriately (page 43, line 1171).

Reviewer #1

Reviewer 1 Comment 1 (Overall impression): I understand that interview studies are usually exploratory in nature. However, the findings from this study (e.g., Increased child anxiety and disconnectedness, deteriorating parental mental health, barriers to maintaining optimal mental wellbeing due to the lack of routine, the lack of social interaction, and the lack of social support) were not surprising nor gave the field significant insights. The results were not intellectually stimulating because all parents around the world have been suffering from the same things in the past 3 years. This has made the results of this study less significant to the ADHD population specifically.

Response: We agree that many of the findings align with the experiences of families not dealing with ADHD. On page 3 (line 73) and on page 4 (line 100) we acknowledge this similarity by stating, “COVID-19 has had an unprecedented impact on the mental and physical wellbeing of individuals all over the world” and “the pandemic has had a substantial impact on the lives of all individuals and families”, respectively. However, our argument is that the magnitude of the pandemic’s effects (e.g., increased child anxiety, deteriorating parental mental health) is greater among families dealing with ADHD and thus it is important to also capture their experiences. This has been supported by other research showing that adolescents with ADHD were more likely to have sustained negative mental health functioning compared to their peers without ADHD even after the stay-at-home orders were lifted (Breaux et al., 2021). Parents of children with special needs have also been found to experience more mood instability (Asbury et al., 2021) and greater challenges overall (Cortese et al., 2020; Thorell et al., 2022) than parents with typically developing children during the pandemic. Exacerbation of pre-existing mental health challenges has also been shown in children with ADHD during the pandemic, further taxing the family system (Palacio-Ortiz et al., 2020). The current study answers a call for more research to capture the impact of chronic stressors on individuals with ADHD given their longer-lasting impacts (Breaux et al., 2021). We appreciate the reviewer highlighting how this was not made clear enough in the rationale of the study, and thus we have added further context from page 4 (line 99) to page 7 (line 152) to underscore the importance of this work in families with ADHD. 

Asbury K, Fox L, Deniz E, Code A, Toseeb U. How is COVID-19 affecting the mental health of children with special educational needs and disabilities and their families?. Journal of autism and developmental disorders. 2021 May;51(5):1772-80.

Breaux R, Dvorsky MR, Marsh NP, Green CD, Cash AR, Shroff DM, Buchen N, Langberg JM, Becker SP. Prospective impact of COVID‐19 on mental health functioning in adolescents with and without ADHD: Protective role of emotion regulation abilities. Journal of Child Psychology and Psychiatry. 2021 Sep;62(9):1132-9.

Cortese S, Asherson P, Sonuga-Barke E, Banaschewski T, Brandeis D, Buitelaar J, Coghill D, Daley D, Danckaerts M, Dittmann RW, Doepfner M. ADHD management during the COVID-19 pandemic: guidance from the European ADHD Guidelines Group. The Lancet Child & Adolescent Health. 2020 Jun 1;4(6):412-4.

Palacio-Ortiz JD, Londoño-Herrera JP, Nanclares-Márquez A, Robledo-Rengifo P, Quintero-Cadavid CP. Psychiatric disorders in children and adolescents during the COVID-19 pandemic. Revista Colombiana de psiquiatria (English ed.). 2020 Oct 1;49(4):279-88.

Thorell LB, Skoglund C, de la Peña AG, Baeyens D, Fuermaier A, Groom MJ, Mammarella IC, Van der Oord S, van den Hoofdakker BJ, Luman M, de Miranda DM. Parental experiences of homeschooling during the COVID-19 pandemic: Differences between seven European countries and between children with and without mental health conditions. European child & adolescent psychiatry. 2022 Apr;31(4):649-61.

Reviewer 1 Comment 2: Given that a lot of countries start to stop implementing social distancing policies, the findings from this study might not be timely enough. The findings might still give the field some insights if the next pandemic has come to us, however, the authors have only mentioned some general suggestions to help ADHD children. For example, providing greater learning support, providing online extra-curricular activities as energy outlets, and enhancing social interactions. These suggestions are true and important for all students, not specifically for ADHD children, making the theme of this paper (i.e., ADHD caregiving) less coherent. Sharing regarding proactive measures to face another pandemic for ADHD kids specifically could be added to the discussion to deepen the discussions.

Response: The pandemic, although unique in many ways, is also an analogue to many of the high-stress environments that children with ADHD are often faced with, such as greater familial discord, greater familial financial stressors, more challenging school experiences, and lower socioeconomic surroundings. Thus, we argue that the insights and suggestions for ways forward may be broadly applicable. However, we appreciate the reviewer advocating for more ADHD-specific measures and have thus included discussion of these on page 31 (line 678-690).

Reviewer 1 Comment 3 (Research rationale and theoretical foundation): Again, I understand that interview studies are exploratory in nature. However, the theoretical foundation driving this research study was too weak. The academic research gap has not been clearly justified or identified either. The introduction was a bit too short, and the literature review in the introduction was too sketchy and descriptive, which might be one of the reasons the authors were not able to derive the research motivation clearly. Without a clear theoretical framework, the academic quality of an article might be threatened. In the introduction, the authors have mentioned the impact of the COVID-19 pandemic and how families have been affected. They have also provided some reasons why they were particularly interested in the ADHD population. However, to me, they were not academically convincing reasons or motivations to conduct this research. Academic quality could be enhanced by theoretical frameworks. The discussion and review of extant literature under a certain theoretical background might help researchers think of real and academically sound reasons to conduct academic research.

Response: We thank the reviewer for their strong advocation for theory-driven research. We originally did not want to include a full detailing of our theoretical framework to avoid unnecessary complexity, as our goal was to mobilize this knowledge not only for academics but also for families in the broader community. We framed the study’s purpose using more layman’s terminology, as has been standard practice in similar work (Sasaki et al., 2020; Shah et al., 2020; Stayridou et al., 2020; Swansburg et al., 2021). However, we appreciate that this could compromise the quality of the article. We have included the theoretical framework driving the study starting on page 4 (line 99) and have also included more background research. 

Sasaki T, Niitsu T, Tachibana M, Takahashi J, Iyo M. The inattentiveness of children with ADHD may worsen during the COVID-19 quarantine.

Shah K, Mann S, Singh R, Bangar R, Kulkarni R. Impact of COVID-19 on the Mental Health of Children and Adolescents. Cureus. 2020 Aug 26;12(8). doi: 10.7759/cureus.10051

Stavridou A, Stergiopoulou AA, Panagouli E, Mesiris G, Thirios A, Mougiakos T, Troupis T, Psaltopoulou T, Tsolia M, Sergentanis TN, Tsitsika A. Psychosocial consequences of COVID‐19 in children, adolescents and young adults: a systematic review. Psychiatry and Clinical Neurosciences. 2020 Nov;74(11):615.

Swansburg R, Hai T, MacMaster FP, Lemay JF. Impact of COVID-19 on lifestyle habits and mental health symptoms in children with attention-deficit/hyperactivity disorder in Canada. Paediatrics & child health. 2021 Aug;26(5):e199-207.

Reviewer 1 Comment 4 (Methodologies and Results): The sampling rationale was unclear. The reason why ADHD kids must be between 7 and 12 years was not clear. The sentence “Living with them for at least some of the time” was ambiguous. The meaning of “at least some of the time” was unclear. Without a clearly written method section, the replicability of this study will be seriously affected. 

Response: We apologize for the confusion in our wording. The child participants were between the ages of 7 and 12 but we did not restrict recruitment to that range. We have removed this information and provided the mean age and standard deviation only in the demographic table to avoid confusion (Table 1). Regarding the meaning of “living with them for at least some of the time”, this wording was chosen to ensure separated parents with shared custody arrangements could also participate despite not living with their child(ren) with ADHD fulltime. We have included a definition of the wording on page 8 (line 172) to make this clear. 

Reviewer 1 Comment 5: I am not sure about the objectives for interviewing two groups of participants (i.e., parents and their ADHD children) in this paper. It makes more sense to me that the authors wanted to compare these two groups of participants on the same topic, so they interviewed them separately. For example, hypothetically, a group of researchers wanted to study the well-being changes of parents of ADHD kids and ASD kids during the COVID-19 pandemic. Therefore, they recruited 30 parents of ADHD kids and ASD kids respectively, and they were asked the same set of questions. By doing the content analysis, a different set of themes might be found in these two categories of parents. However, this study did not do a similar comparison, parents and kids were asked different questions. This made me confused about the reason why that was.

Response: The reviewer has certainly suggested a valuable study to conduct. However, our study was not focused on understanding the differential impact of the pandemic on children versus parental mental wellbeing. Rather, we wanted to gather information about the family system, which includes both the child(ren) and the parents holistically. We interviewed children separately from their parents to offer children more freedom of expression. Parents and children were asked age-appropriate questions that were different, but related, to account for possible differences in communication or self-reflection ability. We thank the reviewer for highlighting how the reasons for our design and question choices should have been made more clear in the Method, and have included these details on page 9 (beginning on line 207). 

Reviewer 1 Comment 6: Table 2 Page 12—the sub-themes might be a bit redundant. For example, the lack of social support may be a reason why parents thought it was difficult to manage parenting duties. Could they be grouped together? The increased parental anxiety might not be a sub-theme, as it was a manifestation of the main theme “deteriorating parental mental health”.

Response: Each subtheme represents distinct information not captured by other subthemes. However, as the reviewer suggests, there are some overlapping topics across some subthemes e.g., difficulty managing parenting duties was in part due to working from home while also caring for children (lack of support). Nonetheless, difficulties managing parenting duties was also distinctly related to parental feelings of guilt and shame. Additionally, the lack of social support subtheme captures information related to lack of childcare options, lack of social interaction, and lack of physical contact. While there are commonalities among some subthemes, they each capture unique information that would not be conducive to grouping. Also, all subthemes are technically a manifestation of the main theme, as this is the deliberate sub-structuring process in the content analysis. The subtheme “increased parental anxiety” provides a more nuanced perspective of what is contributing to the main theme of “deteriorating parental mental health” as it reflects anxiety-specific ways in which participants’ mental health was suffering.

Other minor issues:

Reviewer 1 Comment 6: Page 3 line 66 (referencing style is not consistent).

Response: Thank you for highlighting this oversight. The referencing style has been fixed for consistency.

Reviewer 1 Comment 7: In some sections, the writing styles might be a bit too casual. For example, between page 22 and page 24, when the authors discussed the contributors to declining parental mental health, the authors repeatedly wrote “many parents” (e.g., “many parents had strong support system prior to the pandemic”), would it be better to change to sentences with milder tone “a strong support system might be more available for parents prior to the pandemic”?

Response: In these sections, we were specifically referring to what parents expressed during the current study. Thus, when language such as “many parents had strong support systems prior to the pandemic” was used, we meant “many parents expressed having strong support systems prior to the pandemic”. We have changed the wording accordingly to better reflect that the sentences do not reflect all parents, but rather the parents from the current study. Thank you for this suggestion.

Reviewer 1 Comment 8: Thank you very much for your hard work.

Response: Thank you for your thoughtful feedback and suggestions. The recommended changes have significantly strengthened the paper.

Reviewer 2

Reviewer 2 Comment 1: The authors have chosen an important topic that deserves timely research, discussion, and follow-up. The authors have demonstrated sufficient analysis and discussion of previous studies in the literature review. The authors have conducted thorough analyses. The authors have provided the way forward.

Response: Thank you for the positive feedback!

Reviewer 2 Comment 2: Consider converting this paragraph (line 118) to an objectives section.

Response: We have converted this paragraph to a “Study rationale and theoretical framework” section (page 4, line 99). We have also added a “Research gap and objectives” section on page 6 (line 147).

Reviewer 2 Comment 3: Include a separate research gap section and identify the research gaps.

Response: We have added a “Research gap and objectives” section on page 6 (line 147).

to make clearer the research gaps and objectives of the study.

Reviewer 2 Comment 4: Add response rate. 

Response: We have added the response rate to page 7 (line 169).

Reviewer 2 Comment 5: Any theory guiding the design of interview questions? Were there enough questions?

Response: The Social Ecological Model (SEM) was used to guide the design of interview questions. The SEM is often used in research designed to identify barriers to behaviour by acknowledging the interdependences between intrapersonal, interpersonal, institutional, community and policy factors influencing behaviour. The verbiage “barriers” was used throughout the interview guide, with specific questions targeting the different interrelated levels (e.g., asking about intrapersonal, interpersonal, community factors impacting mental wellbeing for both children and parents). Children were also asked about barriers but using words like “What was the hardest part about the pandemic?”. We have included a description of the role of SEM in the design of the interview questions on page 8 (beginning on line 188). We argue that there were an adequate number of questions because the interviews were semi-structured with many questions being quite broad e.g., “What are the biggest barriers to maintaining optimal mental health for yourself and for your child during the COVID-19 pandemic?”; as a result, there was opportunity to provide both depth and breadth, with all parental interviews lasting a full hour. Fewer questions were asked from children as they had a more challenging time discussing the impact of the pandemic. We discuss the shortcomings of the child interviews in the Limitations section on page 32 (beginning on line 700).

Reviewer 2 Comment 6: Sample size issue: were any attempts made to interview more participants?

Response: Although we did try to contact more families, many did not respond or expressed not having the time to participate. We wanted to limit recruitment to no more than a 4-month period (between October 2020 and January 2021) as the COVID landscape was constantly changing and we wanted participants to be able to reflect about current more so than past life circumstances. This information is included on page 7 (line 169).

---

## [Decision Letter · Decision Letter 1]

29 Nov 2022

PONE-D-22-20962R1The impact of the COVID-19 pandemic on the mental health of families dealing with attention-deficit hyperactivity disorderPLOS ONE

Dear Barbara Fenesi,

Thank you for submitting your manuscript to PLOS ONE. After careful consideration, we feel that it has merit but does not fully meet PLOS ONE’s publication criteria as it currently stands. Therefore, we invite you to submit a revised version of the manuscript that addresses the points raised during the review process.

We look forward to receiving your revised manuscript.

Kind regards,

Tai Ming Wut

Academic Editor

PLOS ONE

Reviewers' comments:

Reviewer's Responses to Questions

**Comments to the Author**

1. If the authors have adequately addressed your comments raised in a previous round of review and you feel that this manuscript is now acceptable for publication, you may indicate that here to bypass the “Comments to the Author” section, enter your conflict of interest statement in the “Confidential to Editor” section, and submit your "Accept" recommendation.

Reviewer #1: (No Response)

Reviewer #2: All comments have been addressed

2. Is the manuscript technically sound, and do the data support the conclusions?

Reviewer #1: Yes

Reviewer #2: Yes

3. Has the statistical analysis been performed appropriately and rigorously? 

Reviewer #1: Yes

Reviewer #2: Yes

4. Have the authors made all data underlying the findings in their manuscript fully available?

Reviewer #1: (No Response)

Reviewer #2: Yes

5. Is the manuscript presented in an intelligible fashion and written in standard English?

Reviewer #1: Yes

Reviewer #2: Yes

6. Review Comments to the Author

Reviewer #1: 1. Thank you very much for adding a theoretical framework to make this paper more academically sound

2. This paper is about ADHD patients in the COVID-19 pandemic. I do believe that an important literature review section is missing in the introduction. Now the introduction includes a) general impact from the COVID-19 pandemic; b) systems theory; c) a very short research gap discussion paragraph. I think the introduction could be made longer. The authors can write a section especially on reviewing literature regarding how ADHD families react to the COVID-19 pandemic or how they are affected by the COVID-19 pandemic. I tried searching “ADHD AND COVID” on Google Scholar, it gave me a lot of interesting relevant results. By reviewing the current findings regarding ADHD and COVID, I am sure that the authors can deepen the introduction section and make their research motivation more strongly justified. To me the research motivation is suggested suddenly in this version of manuscript.

3. Maybe the section “research gap and objectives” should be put before the section “theory and hypothesis”? is it more sensible?

4. The results are not surprising. The authors discuss the findings one by one in the paper. Any synthesis can be made so that more insights that are significant to the field can be suggested?

5. Any theoretical contribution that can be brought by the findings of this paper? For example, how can the findings help us understand the Systems Theory?

6. To me the Systems Theory is added to the introduction deliberately, it just shows up a bit in the introduction, but it has all been forgotten in the discussion. I think it would be better if the authors can relate back to the theoretical framework in the discussion so that theoretical significance of this paper can be discussed

7. While the writing is very good, systematic, logical, and easy to follow in this paper, it also provides us with a lot of valuable information about the ADHD families, however the paper is too “practical” and the theoretical discussion is kind of not strong enough

8. If the Systems Theory does not work well for the whole storytelling, is there any other theory frameworks in other perspectives that can help? For example, theories in family counseling? I am not familiar with them either, just help brainstorm together.

9. Thank you very much for your hard work to make us understand ADHD families in this hard time.

Reviewer #2: (No Response)

7. PLOS authors have the option to publish the peer review history of their article (what does this mean?). If published, this will include your full peer review and any attached files.

Reviewer #1: **Yes: **On-Ting Lo

Reviewer #2: No

---

## [Author Response · Author response to Decision Letter 1]

12 Dec 2022

Reviewer #1

Comment 1 Thank you very much for adding a theoretical framework to make this paper more academically sound.

Response: Thank you for the thoughtful suggestion, we agree that it has made the paper more academically sound. 

Comment 2: This paper is about ADHD patients in the COVID-19 pandemic. I do believe that an important literature review section is missing in the introduction. Now the introduction includes a) general impact from the COVID-19 pandemic; b) systems theory; c) a very short research gap discussion paragraph. I think the introduction could be made longer. The authors can write a section especially on reviewing literature regarding how ADHD families react to the COVID-19 pandemic or how they are affected by the COVID-19 pandemic. I tried searching “ADHD AND COVID” on Google Scholar, it gave me a lot of interesting relevant results. By reviewing the current findings regarding ADHD and COVID, I am sure that the authors can deepen the introduction section and make their research motivation more strongly justified. To me the research motivation is suggested suddenly in this version of manuscript.

Response: We have included a section capturing the recent work that has been done on the impact of COVID-19 among ADHD individuals and families. This begins on line 100.

Comment 3: Maybe the section “research gap and objectives” should be put before the section “theory and hypothesis”? is it more sensible?

Response: Thank you for this suggestion. We have moved “research gap and objectives” before the “study rationale and theoretical framework” section. We agree that this flow is more sensible. 

Comment 4 The results are not surprising. The authors discuss the findings one by one in the paper. Any synthesis can be made so that more insights that are significant to the field can be suggested?

Response: The reporting structure in content analysis is not based on the novelty of findings relative to the literature but based on the frequency of certain themes arising in interviews and on inter-data uniqueness of the themes. While there are commonalities among some themes and subthemes, they each capture unique information that would not be conducive to grouping. Also, each theme/subtheme has an associated and representative quote from participants to offer explicit and nuanced insight into the topics that emerged in the data. We appreciate how the results may seem overly detailed to some, but we strongly advocate for as much detail and transparency in the reporting as possible, even if some of the findings are not novel.

Comment 5: Any theoretical contribution that can be brought by the findings of this paper? For example, how can the findings help us understand the Systems Theory?

Response: We have incorporated discussion of the theoretical contributions on page 31 (line 683).

Comment 6: To me the Systems Theory is added to the introduction deliberately, it just shows up a bit in the introduction, but it has all been forgotten in the discussion. I think it would be better if the authors can relate back to the theoretical framework in the discussion so that theoretical significance of this paper can be discussed.

Response: Thank you for pointing this out. We have incorporated discussion on how the findings relate back to Systems Theory on page 31 (line 683). 

Comment 7: While the writing is very good, systematic, logical, and easy to follow in this paper, it also provides us with a lot of valuable information about the ADHD families, however the paper is too “practical” and the theoretical discussion is kind of not strong enough.

Response: We have incorporated more theoretical discussion starting on pages 31-32. While we agree that discussing more theory is beneficial, and we have gone ahead and done so, a major goal of this work was to offer insight into how families dealing with ADHD experienced COVID-19 and to identify potential responses and practical safeguards for future times of crisis. Thus, it was a deliberate aim of the paper to offer practical insights. 

Comment 8: If the Systems Theory does not work well for the whole storytelling, is there any other theory frameworks in other perspectives that can help? For example, theories in family counseling? I am not familiar with them either, just help brainstorm together.

Response: We have incorporated Attachment theory into the paper (page 8 line 178 of the introduction and page 32 line 703 of the discussion). This theory also offers a valuable framework to contextualize the study. 

Comment 9: Thank you very much for your hard work to make us understand ADHD families in this hard time.

Response: Thank you for your thoughtful feedback and suggestions. The recommended changes have significantly strengthened the paper.

---

## [Decision Letter · Decision Letter 2]

27 Feb 2023

PONE-D-22-20962R2The impact of the COVID-19 pandemic on the mental health of families dealing with attention-deficit hyperactivity disorderPLOS ONE

Dear Barbara Fenesi,

Thank you for submitting your manuscript to PLOS ONE. After careful consideration, we feel that it has merit but does not fully meet PLOS ONE’s publication criteria as it currently stands. Therefore, we invite you to submit a revised version of the manuscript that addresses the points raised during the review process.

We look forward to receiving your revised manuscript.

Kind regards,

Tai Ming Wut

Academic Editor

PLOS ONE

Reviewers' comments:

Reviewer's Responses to Questions

**Comments to the Author**

1. If the authors have adequately addressed your comments raised in a previous round of review and you feel that this manuscript is now acceptable for publication, you may indicate that here to bypass the “Comments to the Author” section, enter your conflict of interest statement in the “Confidential to Editor” section, and submit your "Accept" recommendation.

Reviewer #2: All comments have been addressed

Reviewer #3: (No Response)

2. Is the manuscript technically sound, and do the data support the conclusions?

Reviewer #2: Yes

Reviewer #3: No

3. Has the statistical analysis been performed appropriately and rigorously? 

Reviewer #2: (No Response)

Reviewer #3: No

4. Have the authors made all data underlying the findings in their manuscript fully available?

Reviewer #2: Yes

Reviewer #3: (No Response)

5. Is the manuscript presented in an intelligible fashion and written in standard English?

Reviewer #2: Yes

Reviewer #3: No

6. Review Comments to the Author

Reviewer #2: (No Response)

Reviewer #3: Basically, it is an interesting article and authors showed efforts in looking at the ADHD group during COVID-19. Since it is a qualitative research, the findings should be more in-depth. Would there be any correlation between demographic factors such as family background and educational background with the ADHD child in handling the online teaching or at home? In addition, those incidents also happened in normal family such as anxiety and uncertainty during the pandemic. From the findings, will the three different group of diagnosis needed different level of assistance as no one solution could fit all. Furthermore, COVID-19 has come to an end very soon (WHO), will it be still relevant for addressing the issue.

7. PLOS authors have the option to publish the peer review history of their article (what does this mean?). If published, this will include your full peer review and any attached files.

Reviewer #2: No

Reviewer #3: No

---

## [Author Response · Author response to Decision Letter 2]

1 Mar 2023

Reviewer 3:

Comment 1: Since it is a qualitative research, the findings should be more in-depth. 

Response: We disagree that our findings are not in-depth. We have unpacked each theme and subtheme as it relates to previous research and have situated our results within the context of two key theoretical frameworks. We have also included a detailed section that offers practical suggestions for future clinical practice and research. 

Comment 2: Would there be any correlation between demographic factors such as family background and educational background with the ADHD child in handling the online teaching at home? 

Response: Although this is an interesting question, the suggestion concerns a very specific topic and is beyond the scope of the current research study. To avoid the paper becoming too broad and disorganized we do not think it is appropriate to comment on such niche topics.

Comment 3: In addition, those incidents also happened in normal family such as anxiety and uncertainty during the pandemic. 

Response: We have already addressed throughout the manuscript how “normal” families also experienced many adverse situations and we have emphasized in many places how it is imperative to address how the magnitude of the effects of the pandemic on mental health outcomes is amplified in families dealing with ADHD. 

Comment 4: From the findings, will the three different group diagnosis needed different level of assistance as no one solution could fit all. 

Response: Similar to a comment above, this is an interesting and important point, but it is beyond the scope of the current study. We did not investigate the differences in ADHD diagnoses and whether they would require a different approach. 

Comment 5: Furthermore, COVID-19 has come to an end very soon, will it still relevant for addressing the issue. 

Response: We have already addressed this concern and indicated that families dealing with ADHD often find themselves in high-stress environments, often mimicking similar features of COVID-19 (isolation, reduced access to resources, lack of support, etc.) Thus, we emphasize that the findings from the current study are applicable beyond the parameters of a pandemic.

---

## [Editor Report · Decision Letter 3]

6 Mar 2023

The impact of the COVID-19 pandemic on the mental health of families dealing with attention-deficit hyperactivity disorder

PONE-D-22-20962R3

Dear Barbara Fenesi,

We’re pleased to inform you that your manuscript has been judged scientifically suitable for publication and will be formally accepted for publication once it meets all outstanding technical requirements.

Kind regards,

Tai Ming Wut

Academic Editor

PLOS ONE
---

## [Editor Report · Acceptance letter]

8 Mar 2023

PONE-D-22-20962R3 

The impact of the COVID-19 pandemic on the mental health of families dealing with attention-deficit hyperactivity disorder 

Dear Dr. Fenesi:

I'm pleased to inform you that your manuscript has been deemed suitable for publication in PLOS ONE. Congratulations! Your manuscript is now with our production department. 

Kind regards, 

on behalf of

Dr. Tai Ming Wut 

Academic Editor

PLOS ONE